# AdmTree: Compressing Lengthy Context with Adaptive Semantic Trees

**Yangning Li**[1,2]*, **Shaoshen Chen**[1]*, **Yinghui Li**[1‡], **Yankai Chen**[3],
**Hai-Tao Zheng**[1,2‡], **Hui Wang**[2], **Wenhao Jiang**[4‡], **Philip S. Yu**[3]
[1]Shenzhen International Graduate School, Tsinghua University
[2]Peng Cheng Laboratory       [3]University of Illinois Chicago
[4]Guangdong Laboratory of Artificial Intelligence and Digital Economy (SZ)

## Abstract

The quadratic complexity of self-attention constrains Large Language Models (LLMs) in processing long contexts—a capability essential for many advanced applications. Context compression aims to alleviate this computational bottleneck while retaining critical semantic information. However, existing approaches often fall short: explicit methods may compromise local detail, whereas implicit methods can suffer from positional biases, information degradation, or an inability to capture long-range semantic dependencies. We propose *AdmTree*, a novel framework for adaptive, hierarchical context compression with a central focus on preserving high semantic fidelity while maintaining efficiency. *AdmTree* dynamically segments input based on information density, utilizing gist tokens to summarize variable-length segments as the leaves of a semantic binary tree. This structure, together with a lightweight aggregation mechanism and a frozen backbone LLM (thereby minimizing new trainable parameters), enables efficient hierarchical abstraction of the context. By preserving fine-grained details alongside global semantic coherence, mitigating positional bias, and dynamically adapting to content, *AdmTree* robustly retains the semantic information of long contexts.

## 1 Introduction

Large Language Models (LLMs) [2, 3, 52] have demonstrated remarkable proficiency in processing and understanding long contexts, enabling advances in retrieval-ugmented generation [19, 55] and agentic system [54], etc. However, handling long contexts remains computationally intensive due to the quadratic complexity of self-attention with input token length. This leads to high memory consumption and inference latency. Consequently, **context compression** has emerged as a critical technique, aiming to *reduce input token length while preserving maximal semantic integrity*.

Despite promising results, most existing methods **fail to simultaneously preserve information across multiple semantic dimensions**, such as global versus local semantics, or information across positions. This inability in preserving different information dimensions may leads to poor generalization across real-world tasks, which demand distinct types of semantic information. Specifically, existing methods can be divided into two main categories. ***Explicit methods*** [25, 56, 57] directly shorten text by removing content deemed less essential for overall understanding. Although these methods effectively capture global meaning, they often disrupt local coherence due to excessive omission, leading to the loss of fine-grained details. In contrast, ***implicit methods*** [15, 20, 33, 41] encode long contexts into compact latent vectors (also called "gist tokens") in a flat manner. These methods achieve higher compression ratios, but exhibit different compression efficiency to context at distinct positions. As evidenced by [7, 23, 40], implicit compression methods are prone to positional

---

*Equal Contribution. ‡: Corresponding Author. This work was primarily conducted at GML under the leadership of Wenhao.

39th Conference on Neural Information Processing Systems (NeurIPS 2025).

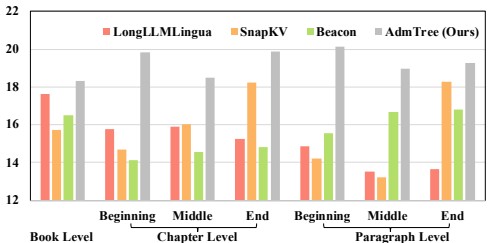
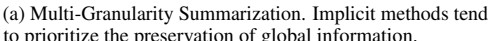
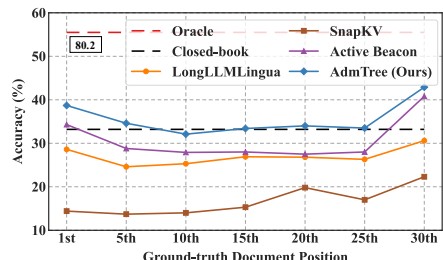

(a) Multi-Granularity Summarization. Implicit methods tend to prioritize the preservation of global information.

(b) Multi-Document Question Answering. Explicit methods exhibit varying compression effectiveness across content at different positions.

Figure 1: Pre-experiments demonstrate that existing methods struggle to balance semantic information of different dimensions for different types of tasks.

bias, often overlooking information from the earlier or middle parts of the context. In other words, less salient information is easily overshadowed by more prominent content.

To mitigate semantic loss caused by position bias, some implicit methods also explored recursive compression [11, 21, 58]. These methods progressively condense segmented contexts into serialized gist tokens. However, they often rely on fixed-size segments without considering variations in information density, leading to imbalanced compression loads across different input regions. Moreover, such linear manner still causes semantic information to degrade progressively during recursive compression, making it difficult to capture long-range dependencies and maintain global coherence.

To overcome these limitations, we draw inspiration from cognitive science, where hierarchical structures are fundamental to human information processing [6, 22]. Hierarchical representations naturally balance *semantic breadth* (coverage across contexts) and *depth* (granularity of detail). Moreover, it establishes *bidirectional connections* among leaf nodes, mitigating the information degradation by previous unidirectional compression. Motivated by this, we introduce the **Ad**aptive Se**m**antic **Tree** Compressor (**AdmTree**), a novel hierarchical context compression framework designed to preserve information integrity more effectively. Unlike methods relying on fixed chunking or flat representations, AdmTree performs compression in a dynamic and structured manner: *(i)* first, long input context is dynamically segmented into variable-length units based on information density. *(ii)* then, gist tokens are interleaved among segments to compress them, serving as the leaf nodes, and a semantic tree is then constructed bottom-up upon these leaves. *(iii)* finally, AdmTree generates responses conditioned on the semantic tree, which can be dynamically updated as new context arrives.

Extensive experiments are conducted to validate the effectiveness of AdmTree. In main experiments, AdmTree consistently achieves state-of-the-art performance across five task types and more than ten datasets from LongBench [4], surpassing baseline methods by over 10% while maintaining **high inference efficiency**. In some QA tasks, AdmTree even outperforms the strongest baseline Beacon [58], by up to 20 points, demonstrating its remarkable ability to **comprehensively preserve semantic information**. Further analysis also confirms that AdmTree exhibits strong **scalability** and efficiency in dynamic context scenarios. Additionally, appendix results highlight the **interpretability** of AdmTree powered by attention over tree nodes, compared to previous black-box implicit compression methods.

## 2 Preliminaries

### 2.1 Task Formulation

Following the instruction tuning setting, we define input context as $\mathbf{X} = (\mathbf{X}^{\text{inst}}, \mathbf{X}^{\text{doc}})$, where $\mathbf{X}^{\text{inst}}$ is task instruction (query) and $\mathbf{X}^{\text{doc}}$ is input long document. The compressed context $\tilde{\mathbf{X}}$ is either text tokens or semantic vectors, based on which LLMs generate responses. The objective of context compression can be formulated as maximize information retention while minimizing redundancy:

$$\max_{\tilde{\mathbf{X}}} I(\tilde{\mathbf{X}}; \mathbf{Y}) - \lambda \mathcal{R}(\widetilde{\mathbf{X}}),$$

in which $I(\widetilde{\mathbf{X}}; \mathbf{Y})$ measures the mutual information between the compressed context $\widetilde{\mathbf{X}}$ and ground-truth answer $\mathbf{Y}$, ensuring that **essential information is preserved**. The second term $\mathcal{R}(\widetilde{\mathbf{X}})$ is a

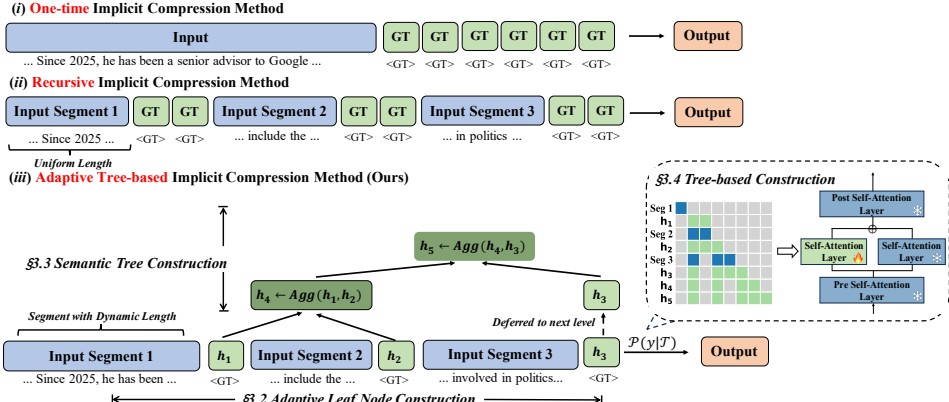

Figure 2: Comparison of compression frameworks. All methods generate responses conditioned on the representation in green. Unlike other methods compress in a linear manner, AdmTree dynamically balances context compression both horizontally and vertically through adaptive gist token allocation and tree hierarchy. Meanwhile, bidirectional aggregation mitigates information degradation.

regularization term that penalizes excessive redundancy, **promoting shorter** $\widetilde{\mathbf{X}}$. The parameter $\lambda$ controls the trade-off between information retention and compression.

To quantify the compression degree, the compression ratio is defined as the proportion of original input length to compressed context length, given by: $\tau = \frac{|\mathbf{X}|}{|\widetilde{\mathbf{X}}|} \geq 1$, where $|\cdot|$ is the token number.

## 2.2 Hard Trade-offs Between Different Information Dimensions in Compression

To preliminarily illustrate the limitations of both explicit and implicit compression methods in balancing semantic information across multiple dimensions, we conduct experiments on two probing tasks: *(i)* **Multi-Granularity Summarization**. Leveraging the BookSum dataset [31], this task requires the model to generate summaries at the book, chapter, and paragraph levels for a given article. It **evaluates the model's capacity to preserve both global and local semantic content**. *(ii)* **Multi-Document Question Answering**. For this task, the NaturalQuestion dataset [32] was employed. The task involves answering questions based on 30 input documents, only one of which contains the ground-truth answer. By placing the ground-truth document at different positions, we verify the model capability to compress information across various locations.

As shown in Figure 1a, we observe that as the summarization granularity becomes finer (from book to paragraph level), the performance of explicit method (LongLLMLingua) consistently declines, while no such trend is observed for implicit methods. This suggests that explicit compression methods tend to prioritize global information while neglecting finer details. In Figure 1b, both Activation Beacon and SnapKV demonstrate better performance when the ground-truth document is placed at the end. This results in a performance curve characterized by an upward trajectory in its latter segment. The "lost in the middle" phenomenon [40] indicates that implicit methods are more prone to positional bias, preferentially retaining information closer to compression tokens. The above preliminary experiments demonstrate the existing compression methods struggle to retain semantic information in multiple dimensions completely, motivating us to propose a more balanced compression framework.

## 3 AdmTree Framework

### 3.1 Overview

As shown in Figure 2, existing one-time or recursive compression methods still operate in a linear or flat manner. They all suffer from information degradation due to position bias. In contrast, AdmTree is a novel framework for adaptive hierarchical context compression. Its dynamic tree structure not only balances semantic preservation in terms of both breadth and depth, but also mitigates the information degradation caused by unidirectional compression through hierarchical aggregation. In general, it dynamically segments input context, assigns gist tokens to summarize each segment, and builds a tree structure to capture contextual information at multiple granularities. This hierarchical representation

is then utilized for compression, with only minimal parameters trained. AdmTree comprises three key stages: dynamic leaf gist token construction, semantic tree construction, tree-based compression.

## 3.2 Adaptive Leaf Gist Token Construction

This stage transforms an input context $\mathbf{X}$ into an alternating sequence of textual sub-segments $\mathbf{s}_k$ and leaf gist tokens $\langle\text{GT}\rangle_k$. The primary objective is to achieve adaptive segmentation based on information density, governed by a compression strategy, to prepare input for hierarchical summarization.

$$\mathbf{X} \xrightarrow{\text{Segmentation \& Budgeting}} \mathbf{X}' = [\mathbf{s}_1, \langle\text{GT}\rangle_1, \mathbf{s}_2, \langle\text{GT}\rangle_2, \ldots, \mathbf{s}_M, \langle\text{GT}\rangle_M]$$

Each leaf gist token $\langle\text{GT}\rangle_k$ is designated to summarize its preceding sub-segment $\mathbf{s}_k$, and the attention scope is restricted to its associated sub-segment and preceding leaf gist tokens. The process involves initial uniform partitioning of $\mathbf{X}$ into segments $\mathbf{X}_i$, followed by adaptive budget reallocation (determining the number of sub-segments $b'_i$ within each $\mathbf{X}_i$) using an information content score (Eq. 2) to define the final sub-segments $\mathbf{s}_k$ and the placement of their corresponding $\langle\text{GT}\rangle_k$. A dedicated token, $\langle\text{GT}\rangle$, is incorporated into the LLM's vocabulary for this purpose.

**Initial Segmentation**    Given an input context $\mathbf{X} = [x_1, \ldots, x_{\mathcal{L}}]$ of length $\mathcal{L}$, AdmTree first partitions it uniformly into initial segments $\mathbf{X_i}$, each of length $n$:

$$\mathbf{X_i} = \left[ x_{(i-1)n+1}, \ldots, x_{\min(i\cdot n, \mathcal{L})} \right], \tag{1}$$

where $n$ is constrained not to exceed the maximum sequence length supported by the LLM.

**Adaptive Compression Budget Allocation**    Recognizing that information density varies across segments, initial uniform segments are adaptively further partitioned: segments with higher information density or complexity are divided into finer sub-segments with more gist tokens (implying a lower local compression ratio).

A simple yet effective strategy is proposed for this budget reallocation. Given the overall compression ratio $\tau$, the initial budget of leaf gist tokens $b_i$ for each segment $\mathbf{X_i}$ (of length $n$) is $n/2\tau$. We first quantify the informational value of each segment $\mathbf{X_i}$ using an entropy-adjusted perplexity score:

$$\text{Score}(\mathbf{X_i}) = \text{PPL}(\mathbf{X_i}) \cdot \exp\left(-\lambda \cdot \text{Entropy}(\mathbf{X_i})\right), \tag{2}$$

where $\text{Entropy}(\cdot)$ measures its intrinsic information content, and $\lambda$ is a hyperparameter balancing PPL and entropy. Segments are then ranked based on this score, and the gist token budgets are redistributed as follows:

$$b'_i = \begin{cases} n/\tau, & \text{if } \mathbf{X_i} \text{ ranks in the top 25\% by Score} \\ n/2\tau, & \text{if } \mathbf{X_i} \text{ ranks in the middle 25\% by Score} \\ n/4\tau, & \text{if } \mathbf{X_i} \text{ ranks in the bottom 50\% by Score} \end{cases} \tag{3}$$

This adaptive allocation strategy still maintains the overall compression ratio $\tau$ globally, while adapting locally to information density. Finally, AdmTree partitions each original segment $\mathbf{X_i}$ into $b'_i$ finer sub-segments, appending one gist token $\langle\text{GT}\rangle$ after each sub-segment. Let $\mathbf{s}_{i,j}$ denote the $j$-th sub-segment of the original segment $\mathbf{X_i}$. The processed segment $\mathbf{X}'_i$ becomes a sequence of sub-segments and their corresponding gist tokens:

$$\mathbf{X}'_i = \left[ \mathbf{s}_{i,1}, \langle\text{GT}\rangle_{i,1}, \mathbf{s}_{i,2}, \langle\text{GT}\rangle_{i,2}, \ldots, \mathbf{s}_{i,b'_i}, \langle\text{GT}\rangle_{i,b'_i} \right], \tag{4}$$

where each sub-segment $\mathbf{s}_{i,j}$ has a length $\alpha_{i,j} = n/b'_i$. These $\langle\text{GT}\rangle_{i,j}$ tokens form the leaf nodes of the semantic tree.

## 3.3 Semantic Tree Construction

Here, we detail the construction of current semantic tree $\mathcal{T}_{\leq k}$ from the set of leaf gist token representations $\{h_{\langle\text{GT}\rangle_1}, \ldots, h_{\langle\text{GT}\rangle_k}\}$. The construction is scalable and incremental: when processing sub-segment $\mathbf{s}_k$, the existing tree $\mathcal{T}_{<k}$ (which summarizes preceding $\langle\text{GT}\rangle_1, \ldots, \langle\text{GT}\rangle_{k-1}$) is partially reused to form $\mathcal{T}_{\leq k}$ by integrating the new gist token's representation $h_{\langle\text{GT}\rangle_k}$ (obtained as described in Sec. 3.4). The general bottom-up aggregation process is:

$$\{h_{\langle\text{GT}\rangle_1}, h_{\langle\text{GT}\rangle_2}, \ldots, h_{\langle\text{GT}\rangle_M}\} \xrightarrow{\text{Hierarchical Aggregation}} \mathcal{T}_M.$$

While the tree structure can be flexible, we employ the binary tree. Meanwhile, to avoid introducing extra consumption, we do not use padding tokens to keep tree balanced. Instead, any remaining unpaired gist token is deferred to the aggregation at next level.

Once the structure is determined, the aggregation function is defined as: $\text{Agg} : \mathcal{P}(\mathbb{R}^d) \to \mathbb{R}^d$, where $\mathcal{P}$ denotes the power set and $\mathbb{R}^d$ is the hidden state dimension. Starting from the leaf gist tokens, the aggregator iteratively combines hidden states $h_u$ for all children $u$ of node $v$ to form the parent node's hidden state $h_v$:

$$h_v = \text{Agg}(\{h_u \mid u \in C_v\}), \tag{5}$$

Here, $C_v$ denotes the set of children of node $v$. For efficiency, the aggregation function $\text{Agg}(\cdot)$ is implemented by a single-layer self-attention mechanism, followed by an averaging operation:

$$\text{Agg}(C_v) = \text{Average}\left(\text{Self-Attn}_{\text{agg}}(\{h_u \mid u \in C_v\}; \theta_{\text{agg}})\right), \tag{6}$$

The trainable parameters size in $\theta_{\text{agg}}$ is significantly smaller than the entire LLM, and are only built once during inference. Therefore, it does not affect the original inference complexity $\mathcal{O}((\mathcal{L}/\tau)^2)$. Furthermore, the self-attention in aggregation mechanism further allows information from subsequent segments to influence preceding ones, mitigating the unidirectional limitations of causal LLMs.

## 3.4 Tree-based Compression

This stage processes each sub-segment $\mathbf{s}_k$ and its associated gist token $\langle \text{GT} \rangle_k$ by leveraging the semantic summary $\mathcal{T}_{<k}$ of preceding content (i.e., tree built from gist tokens $\langle \text{GT} \rangle_1, \ldots, \langle \text{GT} \rangle_{k-1}$). The primary goal is to generate contextualized representations for tokens in $\mathbf{s}_k$ and for $\langle \text{GT} \rangle_k$. Once the representation is computed, the key ($K_{\langle \text{GT} \rangle_k}$) and value ($V_{\langle \text{GT} \rangle_k}$) representations of $\langle \text{GT} \rangle_k$ are also stored to accelerate the construction of subsequent semantic tree (as per Sec. 3.3).

$$(\mathbf{s}_k, \langle \text{GT} \rangle_k), \mathcal{T}_{<k} \xrightarrow{\text{LLM Encoding}} (H_{\mathbf{s}_k}, h_{\langle \text{GT} \rangle_k}, \{K_{\langle \text{GT} \rangle_k}, V_{\langle \text{GT} \rangle_k}\})$$

Here, $H_{\mathbf{s}_k}$ and $h_{\langle \text{GT} \rangle_k}$ denote the output hidden states of text tokens in $\mathbf{s}_k$ and gist token $\langle \text{GT} \rangle$, respectively. Following prior work [21, 41], AdmTree leverages the original LLM to encode text token, while the $\langle \text{GT} \rangle_k$ token and the nodes from $\mathcal{T}_{<k}$ are processed independently with another trainable attention heads.

As illustrated in Figure 2, to distinctly encode gist tokens and regular text tokens, we introduce two separate attention branches before merging them for joint self-attention. Assuming $h_j^l \in \mathbb{R}^d$ is the hidden state of the $j$-th token at layer $l$. The query, key, and value vectors are computed as:

$$q_j^{l+1} = h_j^l W_q^*, \qquad k_j^{l+1} = h_j^l W_k^*, \qquad v_j^{l+1} = h_j^l W_v^*, \tag{7}$$

where placeholder $* \in \{\text{gt}, \text{tt}\}$ distinguishes gist tokens (gt) from text tokens (tt). Projection matrices for text tokens (e.g., $W_q^{tt}$) are derived from the original LLM, while those for gist tokens (e.g., $W_q^{gt}$) are newly introduced and trained. After individually computing, the representations from the preceding tree context ($\mathcal{T}_{<k}$), the current text sub-segment ($\mathbf{s}_k$), and the current gist token ($\langle \text{GT} \rangle_k$) are concatenated back to form a unified sequence. The original order of each token type is maintained. The combined matrices are:

$$Q = [Q_{\text{tt}}, Q_{\text{gt}}], \qquad K = [K_{\mathcal{T}_{<k}}, K_{\text{tt}}, K_{\text{gt}}], \qquad V = [V_{\mathcal{T}_{<k}}, V_{\text{tt}}, V_{\text{gt}}]. \tag{8}$$

Then, the standard self-attention mechanism with a causal mask $M$ is then applied:

$$H^{l+1} = \text{softmax}\left(\frac{QK^T}{\sqrt{d_k}} + M\right)V, \tag{9}$$

where $d_k$ is the dimension of the key vectors. All tokens are encoded with relative positional embeddings for queries and keys. $H$ is actually the concatenation of $H_{\mathbf{s}_k}$ and $h_{\langle \text{GT} \rangle_k}$. Tree nodes from $\mathcal{T}_{<k}$ are flattened in a left-to-right, bottom-to-top order to be integrated into the sequence.

## 3.5 Compression Learning

AdmTree learns to compress context into the semantic nodes of a tree $\mathcal{T}$ and then generate appropriate responses based on them. Similar to general LLMs, AdmTree is optimized via next-token prediction

task. The prediction for token $x_j$ within the given sub-segment $\mathbf{s}_k$ is conditioned on both the dynamically constructed semantic tree $\mathcal{T}_{<k}$ (summarizing all preceding context) and the local intra-sub-segment context $x_{<j}$. The objective is to minimize the negative log-likelihood:

$$\mathcal{L}_{\text{train}} = \min_{\theta_{\text{gt\_attn}}, \theta_{\text{gt\_emb}}, \theta_{\text{agg}}} \mathbb{E}_{\mathbf{X} \sim \mathcal{D}} \left[ - \sum_{\mathbf{s}_k \in \mathbf{X}} \sum_{x_j \in \mathbf{s}_k} \log P\left(x_j \mid \mathcal{T}_{<k}, x_{<j}; \theta_{\text{gt\_attn}}, \theta_{\text{gt\_emb}}, \theta_{\text{agg}}\right) \right], \quad (10)$$

where the outer sum iterates over all sub-segments $\mathbf{s}_k$ derived from a long context input $\mathbf{X}$. The trainable parameters consist of: $\theta_{\text{gt\_attn}}$, for the attention heads processing gist tokens; $\theta_{\text{gt\_emb}}$, for the token embedding of $\langle \text{GT} \rangle$; and $\theta_{\text{agg}}$, for the aggregator parameters. The parameters of the backbone LLM, $\theta_{\text{LLM}}$, remain frozen throughout the training.

## 4 Experiment Details

### 4.1 Experiment Setting

**Baselines** Following prior work [26, 58] on compression, we introduce two main categories of baseline models for processing long contexts:

*(i) Retrieval-based Methods.* These methods employ a retriever to obtain the most relevant segments based on their similarities to the given query (instruction). We utilize one sparse retrieval method **BM25** [48], and two dense retrieval methods, **Sentence-BERT** [47] and **OpenAI Embeddings**[2].

*(ii) Compression-based Methods.* Unlike retrieval-based methods, these methods directly compress the input text into more concise representations. We select **LongLLMLingua** [26] as a representative of explicit compression methods. **AutoCompressor** [11], **ICAE** [21], **Activation Beacon** [58], and **SnapKV** [38] are employed as implicit compression methods. The former three methods recursively compresses long contexts into semantic vectors, enabling preliminary fine-grained compression.

We also fine-tuned the original LLM using the same training data to serve as the strong baseline, denoted as Original LLM-FT.

**Evaluation Benchmark** We evaluate AdmTree's effectiveness and efficiency on **LongBench** [4]. LongBench is a multitask benchmark assessing LLM comprehension of long contexts. The description for other used dataset can be found in Appendix.

**Backbone LLMs** `LlaMA-2-7B-Chat` and `Qwen-2-7B-Instruct` are employed as the backbone LLMs. We use LlaMA-2-7B-Chat because three out of the four compression baselines utilize it.

**Implementation Details** The training details for AdmTree can be found in Appendix. For the compression ratio in main experiments, given LLaMA-2's maximum sequence length of 4K, we apply $\times 2$ compression for 4K–8K contexts, $\times 4$ for 8K–16K, and $\times 8$ for 16K–32K. For Qwen-2, we uniformly set the compression ratio to $\times 4$. The results are averaged over three inference runs.

### 4.2 Main Experiments

We evaluate the AdmTree and other baseline on LongBench. The performance and latency are reported in Table 1, from which we can observe that:

1. **AdmTree significantly surpasses all other baselines.** Specifically, on the Llama-2-7B and Qwen-2-7B models, AdmTree outperforms the current state-of-the-art compression method Activation Beacon, by 10% and 5% in average scores, respectively. The performance gains are particularly striking in QA tasks, where AdmTree achieves improvements exceeding 327.5% in some tasks. Notably, AdmTree achieves these gains without incurring additional latency compared to other recursive compression methods. This performance gain is primarily attributed to AdmTree's adaptive leaf node allocation and hierarchical architecture, which effectively balances semantic information across different dimensions. Meanwhile, the utilization of a single-layer Transformer for the leaf node representation in AdmTree's semantic tree contributes to its computational efficiency. In contrast, AutoCompressor and ICAE exhibit poor performance, primarily due to their fixed number of gist tokens, which hinders their ability to adeptly handle the diverse long-context scenarios in LongBench.

---

[2]We use `text-embedding-ada-002` at OpenAI.

2. **AdmTree's advantages are more pronounced when compared with retrieval-based methods.** While retrieval methods enable processing lengthy context without any modification on original model, this comes at the expense of performance. These methods particularly struggle with complex, multi-hop question answering, since they may fail to retrieve all necessary relevant context. Consequently, there is a marked performance decline on single and multi-document QA compared to AdmTree.

3. **AdmTree is the only compression-based method that consistently outperforms the strong baseline of full fine-tuning on the original LLM (Original LLM-FT) across both backbone models.** We posit that AdmTree's ability to surpass Original LLM-FT stems from its capacity to retain more comprehensive information within its hierarchical structure. When presented with a task query, AdmTree can efficiently "retrieval" and utilize the most relevant information while effectively filtering out contextual noise. Other methods, such as Beacon, do not consistently achieve this, occasionally underperforming Original LLM-FT on specific tasks.

Table 1: Out of domain evaluation on LongBench [4]. Each task type includes at least two datasets. We report the micro-averaged performance across all datasets. †: the results cited from Bai et al. [4], Zhang et al. [58]

| Methods | LongBench | | | | | | Latency | |
| | SingleDoc | MultiDoc | Summ. | FewShot | Code | AVG | Latency | Speedup |
| --- | --- | --- | --- | --- | --- | --- | --- | --- |
| **LLama-2-7B** | | | | | | | | |
| Original LLM† | 24.7 | 22.4 | 24.6 | 63.2 | 57.7 | 37.2 | 6.4 | 4.0× |
| Original LLM-FT† | 34.8 | 27.5 | 23.2 | 61.8 | 57.8 | 39.8 | 25.6 | - |
| *Retrieval-based Methods* | | | | | | | | |
| BM25 | 25.1 | 23.9 | 24.4 | 56.4 | 33.1 | 32.5 | 6.4 | 4.0× |
| SBERT | 17.1 | 15.8 | 23.6 | 53.2 | 36.8 | 28.8 | 6.4 | 4.0× |
| OpenAI | 28.3 | 16.4 | 16.9 | 23.7 | 50.3 | 25.5 | 6.4 | 4.0× |
| *Compression-based Methods* | | | | | | | | |
| AutoCompressor† [11] | 12.9 | 16.4 | 16.3 | 23.8 | 39.4 | 20.5 | 12.8 | 2.0× |
| ICAE† [21] | 19.5 | 19.2 | 19.5 | 24.8 | 27.8 | 21.8 | 10.2 | 2.5× |
| LongLLMLingua† [26] | 21.5 | 18.8 | 21.7 | 49.5 | 53.2 | 31.5 | 8.5 | 3.0× |
| SnapKV† [38] | 24.2 | 22.6 | 16.3 | 60.1 | 57.7 | 34.6 | 8.5 | 3.0× |
| Beacon† [58] | 34.9 | 27.5 | 25.0 | 61.4 | 57.8 | 40.1 | 8.0 | 3.2× |
| **AdmTree (Ours)** | **36.5** | **36.3** | **26.9** | **65.5** | **60.9** | **44.1** | 7.8 | 3.3× |
| **Qwen-2-7B** | | | | | | | | |
| Original LLM† | 38.8 | 37.5 | 26.7 | 70.1 | 60.3 | 45.7 | 23.5 | 1.0× |
| Original LLM-FT† | 41.0 | 40.6 | 26.8 | 68.5 | 66.1 | 47.4 | 23.5 | - |
| *Retrieval-based Methods* | | | | | | | | |
| BM25 | 28.8 | 31.1 | 23.8 | 54.0 | 32.0 | 34.1 | 5.9 | 4.0× |
| SBERT | 18.4 | 22.9 | 21.6 | 50.4 | 34.6 | 28.9 | 5.9 | 4.0× |
| OpenAI | 30.0 | 19.1 | 23.8 | 22.9 | 51.6 | 27.9 | 5.9 | 4.0× |
| *Compression-based Methods* | | | | | | | | |
| LongLLMLingua† | 24.7 | 20.3 | 26.3 | 55.9 | 50.1 | 34.4 | 7.8 | 3.0× |
| SnapKV† | 38.7 | 37.6 | 26.2 | 67.1 | 60.3 | 45.0 | 7.8 | 3.0× |
| Beacon† | 40.5 | 40.3 | 26.8 | 68.4 | 66.4 | 47.2 | 7.3 | 3.2× |
| **AdmTree (Ours)** | **41.6** | **45.9** | **30.2** | **69.9** | **66.8** | **49.7** | 7.0 | 3.4× |

4. Among compression-based methods, **explicit method (LongLLMLingua) and implicit methods (represented by Activation Beacon) demonstrate varied efficacy across different task types.** For instance, LongLLMLingua achieves competitive performance with AdmTree on summarization tasks but significantly lags behind on question answering tasks. This observation aligns with our assertion in the introduction that explicit compression methods may struggle with tasks demanding fine-grained details. Conversely, AdmTree exhibits consistently strong and balanced performance across all task types. As further evidenced by our analyses in Figure 1, AdmTree maintains stable performance across summarization tasks of different granularities and QA tasks with different ground-truth information location. This consistent performance, with lower variance, demonstrating AdmTree's superior capability in preserving semantic integrity across multiple information dimensions, and also suggests the potential of integrating Mixture-of-Experts (MoE) architectures [8] to further specialize compression across diverse task requirements.

## 4.3 Ablation Experiments

To quantify the contributions of each modules to overall performance, we conducted ablation experiments covering different training stages, model structures, and tree construction strategies. As shown in Table 2, both pre-training and instruction fine-tuning enhance the compression learning. Pre-training exerts the larger impact, probably since it learned from the full-sentence contexts, whereas instruction tuning optimized LLMs only on task outputs. Removing the tree structure (while maintaining the compression ratio) resulted in a severe performance drop to 28.5, demonstrating the effectiveness of our proposed tree architecture. When the adaptive leaf gist token allocation was replaced with uniform allocation, indicating that adaptive compression better balances semantic information across segments. Furthermore, removing the parameters from the self-attention layer in the tree aggregation process also led to a notable performance decrease. In general, single-layer self-attention strikes an effective trade-off between performance and efficiency.

**Tree-node retrieval** To further enhance inference efficiency, we experimented with directly retrieving only the top 75% of tree nodes during inference, based on their attention scores relative to

Table 3: Experiments in dialogue scenarios, where context arrives dynamically rather than requiring one-time compression. ShareGPT dataset [12] is used. AdmTree demonstrates notable flexibility.

| Method | 1 Turn (765 tokens) | | | 2 Turn (3006 tokens) | | | 3 Turn (6491 tokens) | | |
|---|---|---|---|---|---|---|---|---|---|
| | PPL↓ | Latency | TFLOPs | PPL↓ | Latency | TFLOPs | PPL↓ | Latency | TFLOPs |
| AutoCompressors | 8.32 | 0.76 | 8.41 | 6.09 | 1.38 | 38.18 | 7.51 | 1.82 | 96.63 |
| ICAE | 7.72 | 0.45 | 8.78 | 6.55 | 0.76 | 40.62 | 8.13 | 1.20 | 92.37 |
| LongLLMLingua | 5.91 | 0.78 | 8.78 | 4.96 | 1.41 | 48.32 | 4.77 | 1.98 | 145.71 |
| SnapKV | 4.91 | 0.32 | 8.30 | 4.20 | 0.69 | 33.97 | 4.63 | 0.83 | 75.45 |
| Beacon | 4.27 | 0.37 | 8.63 | 3.08 | 0.72 | 34.54 | 2.98 | 0.94 | 75.41 |
| **AdmTree (Ours)** | **4.01** | 0.37 | 8.37 | **2.91** | 0.70 | 34.09 | **2.79** | 0.92 | 75.38 |

the last token. The resulting performance loss is marginal: the model still surpasses competitive baselines such as SnapKV on single-document QA. The scalability of AdmTree thus enables further inference-time optimizations under limited computational budgets.

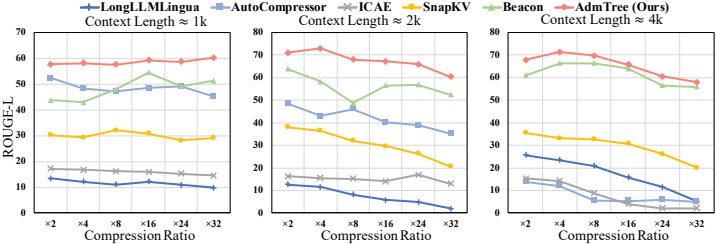

Figure 3: Model performance under varying context lengths and compression ratios.

Table 2: Ablation experiments on LLaMA-based AdmTree.

| Ablation | Single-Doc |
|---|---|
| Full | 36.5 |
| w/o Pre-training | 26.6 |
| w/o Fine-tuning | 29.3 |
| w/o Tree Structure | 28.5 |
| w/o Self-attention | 29.6 |
| w/o Adaptive Leaf Constr. | 34.1 |
| + Tree Nodes Retrieval | 32.8 |

## 5 Analysis Experiments

In this section, several characteristics of AdmTree are investigated, including flexibility, effects across varying context lengths and compression ratios, and compression effects on fine-grained information.

### 5.1 Compression Flexibility

We evaluate the flexibility of compression methods in multi-turn dialogue scenarios, where demands efficient compression of dynamically updated conversational context (utterances). The ShareGPT dataset [12], comprising lengthy dialogues between humans and ChatGPT, was utilized for evaluation. We assessed methods over three consecutive dialogue turns, with the average context length progressively increasing from 765 to 6,491 tokens. Perplexity (PPL) was employed to assess responses quality, while latency and TeraFLOPs (TFLOPs) were reported to measure compression efficiency.

As shown in Table 3, AdmTree consistently achieves the lowest PPL scores across all three dialogue turns. Notably, its PPL decreases as the dialogue progresses and context length increases, which indicates that AdmTree effectively compresses the dialogue history, preserving semantic integrity and conversational structure. Meanwhile, one-time compression methods (e.g., LongLLMLingua and ICAE) incur a dramatic rise in computational cost as the dialogue extends, since they need to the recompress entire context at each turn. While AdmTree and Activation Beacon mitigate this issue as recursive compression methods, which allow for the reuse of compressed representations from previous turns. This scalability is highly desirable for dynamic, real-world applications like online dialogue systems, where context is continuously updated, not merely processed in a single pass.

### 5.2 Compression with Different Context Length and Compression Ratio

Figure 3 presents the performance of various compression methods under different context lengths and compression ratios. Specifically, we use the Multi-Session Conversation (MSC) dataset, constructed by MemGPT [44]. We further process it to obtain dialogue samples with average lengths of 1K, 2K, and 4K tokens. Following MemGPT, ROUGE-L is employed to evaluate response quality. We can found that AdmTree consistently outperforms all baseline models across all context lengths and compression ratios by a substantial margin. At a 1K context length, AdmTree maintains stable performance, even showing slight improvement as the compression ratio increases. For longer contexts, AdmTree exhibits a much smaller degradation in performance compared to other methods.

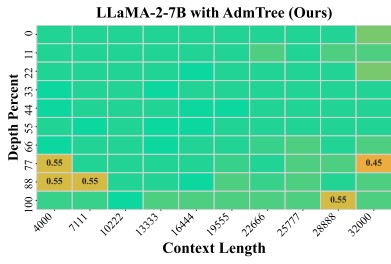
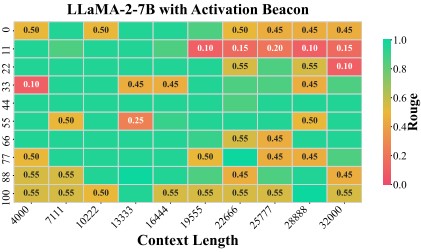

(a) ROUGE scores of responses generated by AdmTree. We show the scores below 0.6.

(b) ROUGE scores of responses generated by Activation Beacon.

Figure 4: Comparison of Needle-in-The-Haystack [27] results with LLaMA-2-7B as backbone LLM. Notably, several baselines collapse under the combination of long context and high compression ratio, with ROUGE-L scores dropping below 10.

## 5.3 Fine-grained Information Compression

We evaluated the fine-grained information retention capabilities of AdmTree and the state-of-the-art Activation Beacon with the challenging Needle-in-the-Haystack dataset. In this task, a specific piece of information ("needle") is embedded at a specific position within a lengthy document ("haystack"), and the model is prompted to retrieve this detail. As illustrated in Figure 4, AdmTree consistently and precisely extracts the target information across documents of varying lengths and needle positions, demonstrating strong semantic preservation. Notably, the length of input contexts in this task substantially exceeds those seen during AdmTree's training, yet the model maintains stable performance.

## 6 Related Work

### 6.1 Compression Methods

As input contexts grow increasingly long, context compression has attracted significant attention. The goal is to reduce input token length while maintaining semantic integrity. The existing methods can be broadly categorized into two categories: (1) Explicit methods that directly shorten text by removing less crucial content based on information theory [25, 26, 37, 45, 51]. These methods range from token-level pruning [25, 45], to sentence-level methods such as CPC [39]. (2) Implicit methods include encoding inputs into soft tokens using methods such as Gisting [41] and Gist-COCO [35], or applying recursive-based compression as in AutoCompressor [11], ICAE [20] and Activation Beacon [58]. Specialized compression methods have also been proposed, including xRAG [10] for retrieval-augmented generation and CCM [29] for online scenarios.

### 6.2 Retrieval Methods

Retrieval methods enable LLMs to process long context by selecting the most relevant segments based on their similarity to a given query. These methods are generally categorized into two types: (1) Sparse Retrieval: use term-frequency-based models, such as TF-IDF [46] and BM25 [48], which represent documents as sparse vectors based on keyword occurrences. (2) Dense Retrieval: employ neural networks to encode queries and documents into continuous vector spaces, enabling semantic similarity matching. Common models include Sentence-BERT [47], Contriever [24], and DPR [28].

## 7 Conclusion

In this paper, to address the limitations of existing context compression methods in preserving semantic fidelity, we propose AdmTree, a novel hierarchical compression framework that dynamically allocates gist tokens and constructs adaptive semantic trees. By integrating information-aware segmentation with lightweight tree-based aggregation, AdmTree comprehensively preserves semantic information across multiple dimensions. Experiments on LongBench show that AdmTree consistently surpasses state-of-the-art baselines in both performance and inference efficiency across diverse tasks. Further analysis also confirms that AdmTree exhibits strong scalability in dynamic context scenarios.

## Acknowledgments

This research is supported by the National Natural Science Foundation of China (Grant No.62276154), the Research Center for Computer Network (Shenzhen) Ministry of Education, the Natural Science Foundation of Guangdong Province (Grant No. 2024TQ08X729 and 2023A1515012914), the Basic Research Fund of Shenzhen City (Grant No.JCYJ20210324120012033, JCYJ20240813112009013, and GJHZ20240218113603006), and the Major Key Project of PCL for Experiments and Applications (PCL2023A09).

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

# A  Additional Related Work

## A.1  Extending LLM Window Size

Unlike the above methods that improve LLM efficiency for long-text processing, some studies focus on expanding the context window of LLMs. A mainstream approach involves progressive pre-training strategies [17, 42, 49], which incrementally extend the maximum context length through phased training. Although these methods achieve strong performance, they incur high training costs and do not reduce inference overhead. In our experiments, we include fine-tuned LLMs trained on the same long-context datasets as strong baselines (Table 1).

# B  Experiment Setting

## B.1  Method Details

**Baselines**    Following prior work [26, 58] on compression, we introduce two main categories of baseline models for processing long contexts:

*(i) Retrieval-based Methods.* These methods employ a retriever to rank segmented long contexts based on their relevance to the given query (instruction). We utilize one sparse retrieval method **BM25** [48], and two dense retrieval methods, **Sentence-BERT** [47] and **OpenAI Embeddings**[3]. After assessing the relevance of the input context to the query using these models, segments with lower relevance are discarded iteratively until the required compression ratio is achieved.

*(ii) Compression-based Methods.* We select **LongLLMLingua** [26] as a representative of explicit compression methods. **AutoCompressor** [11], **ICAE** [21], **Activation Beacon** [58], and **SnapKV** [38] are employed as implicit compression methods. The former three methods recursively compresses long contexts into semantic vectors, enabling preliminary fine-grained compression.

We also fine-tuned the original LLM using the same training data to serve as the baseline, denoted as Original LLM-FT.

**Backbone LLMs**    `LlaMA-2-7B-Chat` and `Qwen-2-7B-Instruct` are employed as the backbone LLMs. We use LlaMA-2-7B-Chat because three out of the four compression baselines utilize it. Throughout the experiments, we adopt LlaMA-2 as the backbone model unless explicitly stated otherwise.

**Implementation Details**    In training AdmTree, we use the same dataset as Zhang et al. [58] to ensure fairness. Specifically, pre-training is conducted on 1B tokens sampled from RedPajama [53]. During fine-tuning, we utilize LongAlpaca [9], BookSum [31], and 16K synthetic samples generated by GPT-3.5 [18]. AdmTree is trained with a batch size of 8 on 8 NVIDIA A800 GPUs, with learning rates set to 5e-5 for pre-training and 1e-5 for fine-tuning, respectively.

For the compression ratio in main experiments, given LLaMA-2's maximum sequence length of 4K, we apply $\times 2$ compression for 4K–8K contexts, $\times 4$ for 8K–16K, and $\times 8$ for 16K–32K. For Qwen-2, we uniformly set the compression ratio to $\times 4$.

## B.2  Dataset Details

**Evaluation Benchmark**    In main experiments, we evaluate AdmTree's effectiveness and efficiency on **LongBench** [4] and **Needle-In-The-Haystack** [27]. LongBench is a multitask benchmark assessing LLMs' comprehension of long contexts across various tasks such as single- and multi-document QA and summarization, with 32K maximum length. Needle-In-The-Haystack assesses the model's ability to extract pertinent fact ("needle") from lengthy context ("haystack").

In other section, we further experiment on diverse benchmarks, including **GSM8K** [14] and **BBH** [50] for demonstration compression, **MSC** [44] and **ShareGPT** [12] for dialogue compression, Arxiv-March23 for summarization scenario, Topic Retrieval for retrieval scenario:

---

[3]We use `text-embedding-ada-002` at OpenAI.

**longbenchv2** This dataset, introduced by Bai et al. [5], spans six major task categories and features documents ranging from 8,000 words to 2 million words in length. The benchmark presents a challenging four-choice question format, with performance evaluated using accuracy as the primary metric.

**GSM8K** This dataset [13] is a widely recognized benchmark for evaluating mathematical reasoning capabilities. It specifically assesses a model's proficiency in performing arithmetic operations and constructing coherent, step-by-step solutions using natural language. In our implementation, we employ a 2-shot learning approach with the original prompt being a complex, multi-step chain-of-thought (CoT) [16] formulation.

**BBH** This dataset [30] is designed for language and symbolic reasoning tasks, with a specific focus on evaluating chain-of-thought (CoT) reasoning capabilities. In our experiments, we employ two distinct CoT prompts as the baseline for comparison.

**ShareGPT** This dialogue dataset [1] is sourced from ShareGPT.com, a platform where users publicly share their ChatGPT interactions across multiple languages and diverse conversational scenarios (e.g., casual chats, writing assistance, and more). For our evaluation, we adopt the curated subset provided by [58] as our test set.

**Arxiv-March23** This dataset is constructed from the most recent academic papers published in the arXiv preprint repository during March 2023. For evaluation purposes, we utilize a curated subset of 500 samples collected as our test set [36]. Input text length is limited to 10,000 characters and we employ DeepSeek to automatically generate summaries for each test instance, which serve as the reference answers [25].

**Topic Retrieval** This evaluation dataset [34] systematically examines model adaptability to varying context lengths and compression requirements. The framework employs structured multi-turn dialog sequences between human users and chatbots.

**MSC** Derived from MemGPT [43], this multi-session benchmark comprises the MSC dataset containing extended human-human dialogues (avg. 2,000 tokens). We systematically transformed the original dataset through prompt engineering to create length variants (1K and 4K tokens) for comprehensive evaluation of context-length robustness.

## C  Analysis Experiments on More Benchmarks and Scenarios

### C.1  Compression on LongBench v2

To ensure consistency with settings in [58], the experiments in main body are conducted on the LongBench benchmark. In Table 4, we further report results on the more challenging LongBench v2 [5], which requiring deeper understanding and reasoning over long contexts. On LongBench v2, AdmTree demonstrates consistent absolute improvements and even more pronounced relative gains. Specifically, it outperforms the state-of-the-art Activation Beacon by +18.5% on LLaMA and +12.7% on Qwen. Moreover, on the newly introduced Long-dialogue History Understanding task, AdmTree brings a notable performance boost, demonstrating its superior ability to preserve dialogue structure and semantics during compression. Besides, compared to the performance on the original LongBench, Activation Beacon no longer consistently surpasses other baselines on some tasks (e.g. Multi-Document QA) in LongBench v2. This trend underscores the potential advantages of tree-structured compression in facilitating structured reasoning over long-context scenarios.

### C.2  Compression on Reasoning Data

Following the experimental setup of Jiang et al. [25], GSM8K and BBH are employed to assess model performance in reasoning and in-context learning scenarios. We introduce a strong baseline, Full-shot, whose prompt lengths on both datasets remain within the maximum context window of LLaMA-2-7B. As shown in Table **??**, AdmTree consistently surpasses all baseline methods. Notably, even under the quarter-shot compression constraint, AdmTree outperforms the Full-shot baseline, achieving

Table 4: Experiment results (%) on LongBench v2 [5]. We also report the micro-averaged performance across all datasets.

| Methods | LongBenchv2 | | | | | | |
| --- | --- | --- | --- | --- | --- | --- | --- |
| | SingleDoc | MultiDoc | ICL | History | Code | Data | **AVG** |
| **LLama-2-7B** | | | | | | | |
| AutoCompressor[11] | 3.1 | 1.5 | 2.6 | 0.8 | 8.9 | 7.2 | 3.1 |
| ICAE [21] | 4.2 | 2.7 | 2.1 | 3.5 | 9.6 | 8.5 | 4.1 |
| LongLLMLingua [26] | 20.0 | 17.6 | 8.6 | 2.6 | 16.0 | 11.8 | 14.9 |
| SnapKV [38] | 18.1 | 15.2 | 23.1 | 16.2 | 18.3 | 19.2 | 18.1 |
| Beacon [58] | 20.0 | 14.4 | 25.9 | 18.0 | 22.0 | 21.2 | 19.5 |
| **AdmTree (Ours)** | **22.9** | **19.2** | **27.2** | **25.6** | **24.0** | **24.2** | **23.1** |
| **Qwen-2-7B** | | | | | | | |
| LongLLMLingua [26] | 22.3 | 19.6 | 12.3 | 8.5 | 13.2 | 20.1 | 18.1 |
| SnapKV [38] | 29.5 | 27.2 | 28.6 | 29.1 | 15.3 | 27.9 | 27.9 |
| Beacon [58] | 34.3 | 18.4 | 29.6 | 30.8 | 16.0 | 33.3 | 28.3 |
| **AdmTree (Ours)** | **34.9** | **25.6** | **33.3** | **38.5** | **22.0** | **33.3** | **31.9** |

Table 5: Experiment results (Exact Match, EM) under different compression ratios on the GSM8K mathematical reasoning [14] and Big-bench Hard (BBH) datasets [50].

| Method | GSM8K | | | BBH | | |
| --- | --- | --- | --- | --- | --- | --- |
| | 1-st. const. ($\sim$5 x) | half-st. const. ($\sim$14 x) | quarter-st. const. ($\sim$20 x) | 1-st. const. ($\sim$3 x) | half-st. const. ($\sim$5 x) | quarter-st. const. ($\sim$7 x) |
| AutoCompressor | 22.7 | 21.5 | 21.2 | 13.71 | 11.56 | 11.23 |
| ICAE | 37.76 | 35.21 | 33.12 | 21.67 | 20.12 | 19.83 |
| Snapkv | 44.56 | 42.95 | 43.81 | 38.61 | 36.83 | 36.01 |
| LongLLMLingua | 42.55 | 43.46 | 43.23 | 33.37 | 39.42 | 38.81 |
| Beacon | 45.74 | 44.36 | 44.13 | 43.95 | 44.25 | 42.44 |
| **AdmTree (Ours)** | **48.12** | **47.93** | **46.85** | **45.06** | **49.7** | **53.53** |
| Zero-shot | 41.87 | | | 36.49 | | |
| Full-shot | 46.88 | | | 43.34 | | |

compression ratios exceeding $20\times$ and $7\times$ on GSM8K and BBH, respectively. These results indicate that AdmTree effectively retains the reasoning process encoded in the original prompts. Interestingly, on the BBH dataset, higher compression ratios yield performance improvements for AdmTree. We attribute this to the relatively short full-shot prompts in BBH (fewer than 500 tokens), where fewer but more compact compressed tokens may enhance compression quality and thereby improve overall performance.

### C.3    Compression on Summarization Scenario

We evaluate the performance of different compression models on the summarization task using an additional dataset, ArXiv-March23. As shown in Table 6, we compare the models under $2\times$ and $4\times$ compression ratios. Across all metrics, AdmTree consistently achieves strong performance. Besides, its performance under the $4\times$ compression ratio does not show significant degradation compared to the $2\times$ setting. In contrast, Activation Beacon performs noticeably worse on the BLEU metric. This may be attributed to the nature of BLEU, which emphasizes n-gram continuity, unlike ROUGE, which focuses on longest common subsequences. The lower BLEU score potentially suggests that Activation Beacon may generate less coherent summaries, potentially introducing unwanted or disjointed tokens.

### C.4    Compression on Retrieval Scenario

We conduct additional experiments in a retrieval scenario using the Topic Retrieval dataset [] as a supplement to the Needle-in-a-Haystack experiment. Different from Needle-in-a-Haystack, which varies the position of the "needle" within a fixed-length context, Topic Retrieval varies the context length by modifying the number of total topics and then querying information about the first topic.

Table 6: Compression performance in summarization scenarios. The ArXiv-March23 dataset is used for evaluation. Metrics include ROUGE-1, ROUGE-2, ROUGE-L, BERTScore, and BLEU.

| Methods | ROUGE-1 | ROUGE-2 | ROUGE-L | BS F1 | BLEU |
|---|---|---|---|---|---|
| Original LLM | 40.35 | 16.22 | 27.22 | 81.57 | 34.88 |
| *2× Compression Constraint* | | | | | |
| AutoCompressor | 9.05 | 0.24 | 6.31 | 12.43 | 5.05 |
| ICAE | 12.87 | 1.03 | 8.42 | 15.25 | 15.77 |
| LongLLMLingua | 31.78 | 9.29 | 20.01 | 74.24 | 24.81 |
| SnapKV | 33.23 | 12.56 | 25.1 | 73.09 | 30.23 |
| Beacon | 35.42 | 15.47 | 25.91 | 77.69 | 16.05 |
| **AdmTree (Ours)** | **36.10** | **15.69** | **26.60** | **79.83** | **35.99** |
| *4× Compression Constraint* | | | | | |
| AutoCompressor | 7.69 | 0.21 | 5.21 | 11.36 | 4.21 |
| ICAE | 10.56 | 0.89 | 7.65 | 13.21 | 13.52 |
| LongLLMLingua | 29.3 | 7.43 | 18.2 | 69.04 | 26.36 |
| SnapKV | 31.25 | 8.92 | 20.53 | 70.21 | 23.21 |
| Beacon | 32.18 | 14.12 | 23.96 | 74.25 | 17.22 |
| **AdmTree (Ours)** | **35.51** | **15.06** | **25.84** | **79.97** | **34.82** |

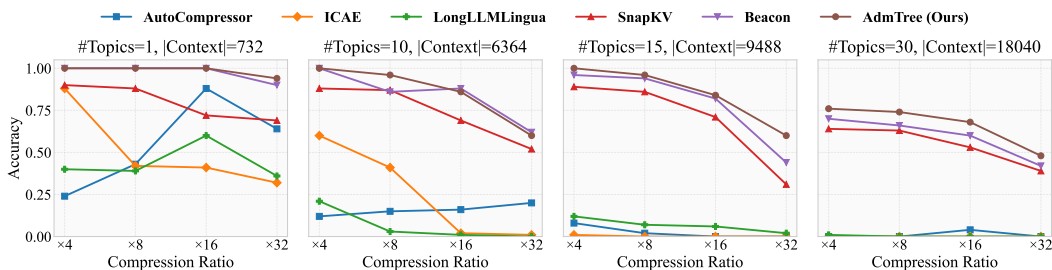

Figure 5: Retrieval accuracy comparison on the Topic Retrieval dataset under different context lengths and compression ratios.

Accuracy is used to measured the retrieval effectiveness after compression. Figure 5 presents the retrieval accuracy of different methods under varying compression ratios and context lengths. AdmTree consistently achieves the best performance across all conditions. Notably, at a shorter context length (Context Length = 732), AdmTree maintains near-lossless compression even at a 16 × compression ratio. For longer contexts, AdmTree also exhibits lower performance degradation compared to other methods. In contrast, AutoCompressor and LongLLMLingua fail to handle context lengths beyond 6000 tokens effectively.

# D   Visualization Experiments

## D.1   Attention Pattern Analysis

Figure 6 presents a KL divergence analysis of the attention distributions for tasks requiring varied semantic information, comparing AdmTree and Activation Beacon. Specifically, we measure the attention scores from the sentence's final token to the preceding compressed (gist) tokens at the onset of generation. Tasks 0–9 correspond to repetition tasks involving different semantic requirements, including the repetition of content from different positions (beginning, middle, end) and at varying granularities (word, sentence, paragraph). AdmTree demonstrates substantially greater diversity in its attention distributions across tasks, indicating that its compressed tokens encode distinct and semantically differentiated information. This diversity likely contributes to its improved performance by enabling more task-specific representation retrieval. Conversely, recursive compression methods such as Activation Beacon not only suffer from semantic degradation through iterative compression,

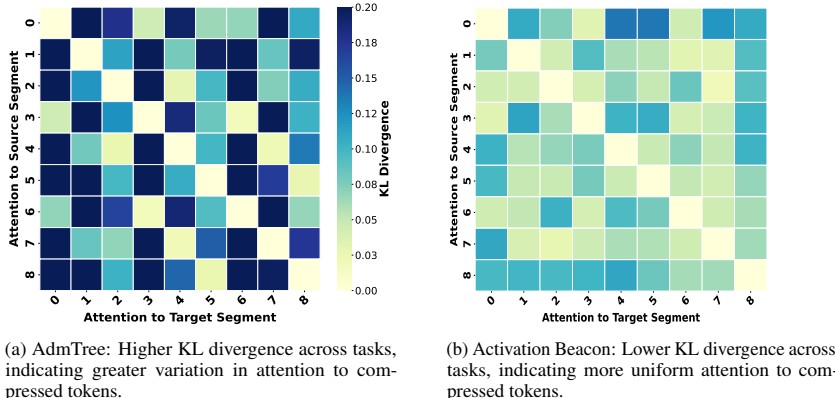

(a) AdmTree: Higher KL divergence across tasks, indicating greater variation in attention to compressed tokens.

(b) Activation Beacon: Lower KL divergence across tasks, indicating more uniform attention to compressed tokens.

Figure 6: The Kullback–Leibler (KL) divergence of the attention distribution from the final token to the compressed tokens across tasks of varying semantic types.

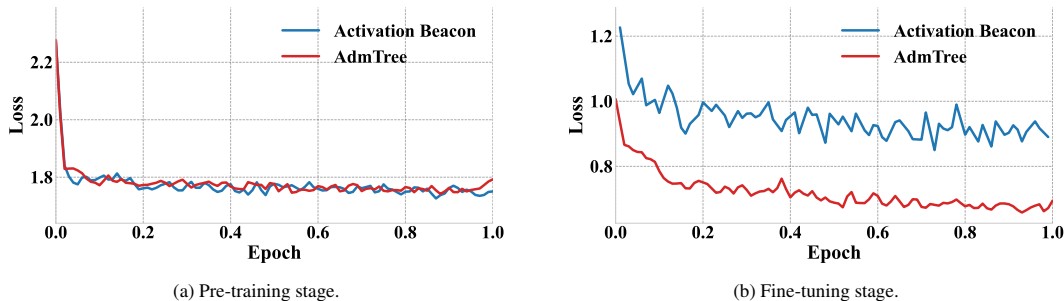

(a) Pre-training stage.

(b) Fine-tuning stage.

Figure 7: Loss curve during the different training stage of AdmTree and Activation Beacon.

but also tend to homogenize the semantic content across compressed tokens, thereby limiting their generalizability across diverse tasks.

## D.2  Training Trends Comparison

AdmTree was trained on the same dataset as Activation Beacon. Figure 7 illustrates the loss convergence trends during both the pre-training and fine-tuning stages for the two methods. As shown, AdmTree achieves better convergence in the multi-task fine-tuning stage.

## D.3  Case Study

We introduce the repetition task, which requires models to faithfully reproduce the original input text without omission or alteration. As shown in Table 7, this task provides an intuitive perspective through which we can observe how different methods preserve semantic information. Compared to the baseline model, our AdmTree achieves near-perfect reconstruction of the original text in both cases, merely leveraging the gist tokens within its semantic tree structure. In contrast, Activation Beacon exhibits severe hallucinations. In the first case, it fabricates content that does not exist in the original text based on its internal knowledge; in the second case, it falls into uncontrolled repetition. These qualitative comparisons clearly demonstrate that AdmTree is more capable of preserving the original semantics in a faithful and complete manner.

Table 7: Case studies on the repetition task. We sample several contexts from the GovReport subset of LongBench and evaluate whether models can accurately faithfully repeat the original content. We compare the outputs of our AdmTree and Activation Beacon. Content in `red` indicates hallucinated or redundant content that deviates from the original text.

| Original Context | Activation Beacon | AdmTree (Ours) |
|---|---|---|
| Colombia, one of the oldest democracies in the Western Hemisphere and the third most populous Latin American country, has endured a multisided civil conflict for more than five decades until President Juan Santos declared the conflict over in August 2017 at the end of a U.N.-monitored disarmament. According to the National Center for Historical Memory 2013 report, presented to the Colombian government as part of the peace process to end the fighting, some 220,000 Colombians died in the armed conflict through 2012, 81% of them civilians. The report also provided statistics quantifying the scale of the conflict, which has taken a huge toll on Colombian society: more than 23,000 selective assassinations between 1981 and 2012; internal displacement of more than 5 million Colombians due to land seizure and violence; 27,000 kidnappings between 1970 and 2010; and 11,000 deaths or amputees from anti-personnel land mines laid primarily by Colombia's main insurgent guerrilla group, the Revolutionary Armed Forces of Colombia (FARC). | Colombia is a founding member of the Pacific Alliance, a trade bloc that seeks to deepen trade integration and investment with its partner countries and reduce barriers to bilateral trade between the four countries. The Pacific Alliance aims to create a common market, allowing for the free movement of businesses and people, and to serve as an export platform to the Asia-Pacific region. Colombia's leadership role in the Pacific Alliance and its accession to the Organization for Economic Cooperation and Development (OECD) in May 2018, following a review of its macroeconomic policies, are new developments. The record expansion of Colombia's coca crop and increasing cocaine exports to the United States, however, may significantly hinder the effort to consolidate peace in Colombia and could potentially increase corruption and extortion. A significant portion of the Colombian public remains skeptical of the peace process and the FARC 's role in Colombia's democracy. Other | Colombia, one of the oldest democracies in the Western Hemisphere and the third most populous Latin American country, has endured a multisided civil conflict for more than five decades until President Juan Ma nuel Santos declared the conflict over in August 2017 at the end of a U.N.-monitored disarmament. According to the National Center for Historical Memory 2013 report, presented to the Colombian government as part of the peace process to end the fighting, some 220,000 Colombians died in the armed conflict through 2012, 81% of them civilians. The report also provided statistics quantifying the scale of the conflict, which has taken a huge toll on Colombian society: more than 23,000 selective assassinations between 1981 and 2012; internal displacement of more than 5 million Colombians due to land seizure and violence; 27,000 kidnappings between 1970 and 2010; and 11,000 deaths or amputees from anti-personnel land mines laid primarily by Colombia's main insurgent guerrilla group, the Revolutionary Armed Forces of Colombia (FARC). |
| The Navy's proposed FY2019 budget requests a total of $2,347 million (i.e., about $2.3 billion) in procurement funding for the CVN-78 program. Congress's decisions on the CVN-78 program could substantially affect Navy capabilities and funding requirements and the shipbuilding industrial base. The Navy's FY2020 budget submission also proposed to not fund the mid-life nuclear refueling overhaul (called a Refueling Complex Overhaul, or RCOH) for the aircraft carrier CVN-75 ( Harry S. Truman ), and to instead retire the ship around FY2024 and also deactivate one of the Navy's carrier air wings at about the same time. On April 30, 2019, however, the Administration announced that it was effectively withdrawing this proposal from the Navy's FY2020 budget submission. The Administration now supports funding the CVN-75 RCOH and keeping CVN-75 (and by implication its associated air wing) in service past FY2024. For additional discussion of this withdrawn budget proposal, see Appendix A . | The Navy's to perform the test on the Navy's to perform the test on the Navy's to perform the test on the Navy's to perform the Navy's to perform the Navy's to perform the Navy's to perform the Navy's to perform the Navy's to perform the Navy's to perform the Navy's to perform the Navy's to perform the Navy's to perform the Navy's to perform the Navy's to perform the Navy's to perform the Navy's to perform the Navy's to perform the Navy's to perform the Navy's to perform the Navy's to perform the Navy's to perform the Navy's to perform the Navy's to perform the Navy's to perform the Navy's to perform the Navy's to perform the Navy's to perform the Navy's to perform the Navy's to perform the Navy's to perform the Navy's to perform the Navy's to perform the Navy's to perform the Navy's to perform the Navy's to perform the Navy's to perform the Navy's to perform the Navy's to perform the Navy's to perform the Navy's to perform the Navy's to perform the Navy's to the Navy's new aircraft carrier, the USS Gerald R. Ford, is scheduled to be delivered to the Navy in 2017. The Navy has been working to complete the ship's initial sea trials, which are scheduled to begin in May 2017. | The Navy's proposed FY2019 budget requests a total $2,347 million (i.e., about $2.3 billion) in procurement funding for the CVN-78 class shipbuilding program. Congress's decisions on the CVN-78 class program and other Navy shipbuilding programs may have significant substantially effects on the Navy's capabilities and funding requirements and the shipbuilding industrial base. The Navy's FY2020 budget submission also proposed not to fund the mid-life nuclear refueling overhaul (called a Refueling Complex Overhaul, or RCOH), for the aircraft carrier CVN-75 (Harry S. Truman), and to instead retire the ship around FY2024 and also deactivate one of the Navy's carrier air wings at about the same time. On April 30, 2019, however, the Administration announced that it was effectively withdrawing this proposal from the Navy's FY2020 budget submission. The Administration now supports funding the CVN-75 RCOH and keeping CVN-75 (and by implication its associated air wing) in service past FY2024. For additional discussion of this withdrawn budget proposal, see Appendix A. |

