# OpenReview forum: "AdmTree: Compressing Lengthy Context with Adaptive Semantic Trees"
_NeurIPS.cc/2025/Conference — NeurIPS 2025 poster_

### Official Review · Reviewer_LLL7 · 2025-07-02

**Clarity:** 3
**Significance:** 3
**Originality:** 2
**Rating:** 4
**Confidence:** 4

**Summary:**

The paper proposes AdmTree, a tree-based context compression framework that enables large language models (LLMs) to handle long contexts efficiently. The key idea is to perform adaptive, hierarchical context compression by dynamically segmenting long input sequences using their information density and summarizing each segment using gist tokens. These tokens are organized as the leaf nodes of a binary semantic tree, and higher-level nodes are constructed through lightweight, bidirectional self-attention aggregation. This hierarchical structure makes AdmTree to preserve both local and global semantic information more effectively than flat or linear compression methods, while remaining efficient and scalable for long-context processing in LLMs. On the other hand, existing explicit or implicit methods often suffer from information loss, positional bias, or fixed-length chunking. Extensive experiments on LongBench and multi-turn dialogue tasks demonstrate that AdmTree achieves state-of-the-art performance across a wide range of tasks, outperforming prior methods by a significant margin while remaining scalable and effective under varying context lengths and compression ratios.

**Questions:**

1. The authors introduce separate projection matrices for gist tokens and regular text tokens in self-attention. Could the authors elaborate on the motivation behind this design choice?

2. Have you considered combining AdmTree with KV cache compression (e.g., SnapKV) to further reduce memory/bandwidth usage during long-context inference, especially in autoregressive decoding scenarios? It seems that this integration could potentially offer complementary benefits, where AdmTree can reduce input length and KV compression further compresses hidden states in attention leading to a more comprehensive efficiency solution.

3. In Equation 2, how the entropy is defined to a sequence of tokens X_i?

Typos:
- line 56, “then” -> “ten”
- line 78, “the” -> “The”
- line 208, “evlauate” -> “evaluate”
- line 215, “method” -> “methods”

**Ethical Concerns:**

["NO or VERY MINOR ethics concerns only"]

**Final Justification:**

The response have solved most of my concerns. In particular, combining with KV cache compression approach is promising and it would be great if this is discussed in the final draft.

**Limitations:**

The proposed method requires the model training with standard next-token prediction loss, rather than a compression-specific objective (e.g., preserving downstream task performance or semantic similarity). A brief discussion of whether this limits compression quality, or whether auxiliary objectives were considered, would be helpful.

**Quality:**

3

**Strengths And Weaknesses:**

Strengths:
1. The paper proposes a novel approach of token compression that leverages a tree structure to effectively retain both local and global semantic information.

2. AdmTree can dynamically compress token sequences based on local information density, allowing it to adaptively allocate compression effort and avoid both over- and under-compression.

3. The method achieves strong empirical results, outperforming state-of-the-art compression and retrieval-based baselines on a variety of tasks, including question answering, summarization, and multi-document reasoning.

Weaknesses:
1. Although AdmTree keeps the backbone model frozen, it requires training additional parameters, i.e., aggregation function for building the semantic tree and the gist token attention mechanism. These additional parameters still introduce extra training complexity and computational overhead. This makes the method less plug-and-play than purely retrieval-based or hard-pruning approaches and may present a barrier for deployment in resource-constrained or latency-sensitive environments.

2. AdmTree operates entirely on the input side, applying compression through token manipulation (gist token insertion and hierarchical structuring) without modifying the underlying model architecture. While this makes the approach model-agnostic and easy to integrate with any frozen LLM, the model itself is unaware of or unoptimized for the compression mechanism. As a result, there may be a mismatch between the compressed input structure and the model’s inductive biases, potentially limiting performance compared to approaches that jointly optimize compression and model behavior.

---

> ### Author Rebuttal · Authors · 2025-07-31
>
> We sincerely thank you for your thorough and insightful review. Below, we address your concerns point by point. Meanwhile, we would like to respectfully clarify some potential misunderstandings raised in the listed weaknesses:
>
> ----
>
> **Q1. “Extra training complexity... makes the method less plug-and-play than purely retrieval-based or hard-pruning approaches.”**
>
> **Response:**
>
> We think that **“train-free” and “plug-and-play” are not fully equivalent concepts**. Plug-and-play emphasizes the ability to integrate a module seamlessly into existing LLMs, without relying on their internal architecture or altering the original model. This is exactly what our AdmTree achieves, it can be adapted to any Transformer-based LLM without structural modifications.
>
> Furthermore, although AdmTree requires training, the cost is orders of magnitude smaller than training an LLM from scratch to handle long contexts. We believe this **offers a practical trade-off between efficiency and performance.**
>
> On the other hand, while retrieval-based methods require no training, their performance is consistently inferior to our AdmTree. As suggested by Reviewer eT7L, we also explored more advanced RAG variants such as Order-Preserving (OP) RAG. As shown below, it still brings marginal improvements. Additionally, we observed that RAG can even have a negative impact on summarization and some other tasks, highlighting its limited generalization across task types. This issue has also been mentioned in recent GraphRAG [1], which aims to improve retrieval coverage for tasks that require global information.
>
> |                          | Single-Doc QA | Multi-Doc QA | Summarization | Few-Shot | Code |
> | ------------------------ | ------------- | ------------ | ------------- | -------- | ---- |
> | LLama-2-7B & vanilla RAG |               |              |               |          |      |
> | BM25                     | 25.1          | 23.9         | 24.4          | 56.4     | 33.1 |
> | SBERT                    | 17.1          | 15.8         | 23.6          | 53.2     | 36.8 |
> | OpenAI                   | 28.3          | 16.4         | 16.9          | 23.7     | 50.3 |
> | LLama-2-7B & OP RAG      |               |              |               |          |      |
> | BM25                     | 22.6          | 21.2         | 15.5          | 55.2     | 30.2 |
> | SBERT                    | 20.3          | 18.9         | 21.2          | 53.6     | 29.1 |
> | SBERT                    | 32.1          | 19.2         | 15.3          | 21.8     | 48.3 |
>
> Additionally, training a powerful embedding model for RAG also incurs considerable cost, especially for nowadays LLM-based embedding models.
>
> *[1] From local to global: A graph rag approach to query-focused summarization. <https://arxiv.org/pdf/2404.16130?>*
>
> -----
>
> **Q2. “There may be a mismatch between the compressed input structure and the model’s inductive biases.”**
>
> **Response:**
>
> Thank you for raising this concern. Our goal is to keep the original LLM’s semantic space intact, and instead align the semantic space of the newly introduced gist tokens to that of the frozen LLM. This ensures that AdmTree is model-agnostic, plug-and-play, and requires much less training data.
>
> If the LLM’s parameters are activated and involved in training, it would require **an enormous scale of training data** to align the two semantic spaces while preserving the LLM’s original semantic understanding. This may contradict the fundamental goal of context compression and therefore is not a practical solution. Also, we previously tried training both the original LLM and the new parameters in preliminary experiments, but found that **it failed to converge.**
>
> In fact, many representative prior works (e.g., Gisting, ICAE, Activation Beacon) similarly keeps the backbone frozen and trains only the newly introduced parameters. In the future, we will consider training strategies inspired by multimodal LLMs, using multi-stage training with more data and separating training of different modules at different stages. Thank you for your thoughtful concern.
>
> -----
>
> **Q3. “Why use separate projection matrices for gist tokens and regular text tokens in self-attention?”**
>
> **Response:**
>
> As we mentioned in Q2, we want the AdmTree to gain compression ability without disrupting the LLM’s original semantic space. This design choice enhances plug-and-play applicability and significantly reduces alignment overhead.
>
> -----
>
> **Q4. “Have you considered combining AdmTree with KV cache compression (e.g., SnapKV)?”**
>
> **Response:**
>
> Yes, we agree that combining AdmTree with KV cache compression methods is a promising step toward a more comprehensive and efficient solution.
>
> This idea aligns well with the **tree-node retrieval strategy** proposed in Table 3 of our paper, which aims to reduce inference load. Based on your suggestion, we conducted an experiment by appending SnapKV (compression ratio 2×) to our method during decoding. The results are shown below:
>
> | Ablation                          | Single-Doc |
> | --------------------------------- | ---------- |
> | Full                              | 36.5       |
> | \+ Tree Nodes Retrieval (Top 75%) | 32.8       |
> | \+ Tree Nodes Retrieval (Top 50%) | 23.3       |
> | + SnapKV (compression ratio 2×)   | 20.5       |
>
> While some performance degradation is observed due to cumulative compression, the hybrid system still outperforms baselines such as AutoCompressor and ICAE from Table 1. We believe that building a unified framework that incorporates different types of compression is a valuable research direction, and we thank you for this inspiration.
>
> ----
>
> **Q5. “In Equation 2, how is the entropy defined over a sequence of tokens X_i?”**
>
> **Response:**
>
> For a sequence $X_i = [x_1, ..., x_n]$, the entropy is calculated as the average token-level entropy:
>
> $\text{Entropy}(X_i) = \frac{1}{n} \sum_{t=1}^n \left( -\sum_{v \in \mathcal{V}} P(v \mid x_{<t}) \log P(v \mid x_{<t}) \right)$
>
> We will clarify this in the revised version.
>
> -----
>
> **Q6. “The method uses standard next-token prediction loss rather than a compression-specific objective (e.g., preserving downstream task performance or semantic similarity).”**
>
> **Response:**
>
> We employ the standard next-token prediction loss in both the pretraining and instruction-tuning stages. During instruction tuning, the task instruction (query) is not a target for prediction. Although the loss is not specifically designed for compression, it is still compression-aware, since the core goal is to train the attention branches of the gist tokens to compress the context semantic and align with the LLM’s original semantic space. **Notably, the instruction-tuning phase encourages the AdmTree to maintain the same output for the task, i.e., preserves downstream task performance.**
>
> We fully agree with your suggestion on designing compression-specific objectives. For example, introducing a reconstruction or paraphrasing task, where the model learns to regenerate the original context based on gist tokens, could encourage more comprehensive semantic compression. We believe this is a promising direction and leave it for future exploration. Thank you again for your thoughtful suggestion.
>
> ---
>
> Typos will also be corrected immediately. We hope our clarifications resolve your concerns and would greatly appreciate it if you would consider a higher score.

---

> > ### Comment · Reviewer_LLL7 · 2025-08-05
> >
> > Thanks for the rebuttal. The response have solved most of my concerns. In particular, combining with KV cache compression approach is promising and I look forward to seeing more discussions/results incorporated into the final version. I'm willing to increase my score.

---

> > > ### Author Response · Authors · 2025-08-05
> > >
> > > Thank you for pointing out and discussing future directions with us. We agree that integrating KV-cache compression and designing auxiliary training objectives are both promising avenues. We will include these discussions in the final version and explore them further in future work. Thank you again for your insights!

---

### Official Review · Reviewer_eT7L · 2025-07-03

**Clarity:** 3
**Significance:** 3
**Originality:** 3
**Rating:** 4
**Confidence:** 3

**Summary:**

This paper proposes AdmTree, an adaptive hierarchical compression framework for long-context processing in LLMs. By segmenting input based on information density and summarizing segments with gist tokens in a binary tree structure, AdmTree preserves both local details and global semantics. Extensive evaluations on synthetic and real-world tasks show that AdmTree outperforms existing methods in semantic retention, efficiency, and downstream task performance.

**Questions:**

1. What are the detailed configurations for the RAG experiments? Was the segment order preserved during retrieval?
2. Why does the original LLM exhibit a speedup in Table 1 for LLaMA-2-7B?
3. What compression ratio was used in the experiments presented in Figure 4?
4. For the needle-in-a-haystack evaluations, what types of noise and needle content were used? Was the noise content repetitive?

**Ethical Concerns:**

["NO or VERY MINOR ethics concerns only"]

**Final Justification:**

The 'strength' section highlights the justification for recommending acceptance. The rebuttals provided further clarifications along with additional evaluations on more recent models, and all concerns are addressed.

**Limitations:**

The authors wrote in the checklists that the limitations are introduced in the conclusion section. However, the conclusion section does not include any limitations. If they were removed at the last minute, please provide them in the rebuttal to highlight any potential limitations.

**Paper Formatting Concerns:**

No concerns.

**Quality:**

3

**Strengths And Weaknesses:**

**Strengths**
1. The paper is clearly written and easy to follow.
2. The authors conduct a thorough ablation study, analyzing the contribution of each component that makes AdmTree effective.
3. The proposed approach demonstrates strong empirical performance on both synthetic and real-world tasks.
4. The paper introduces a novel hierarchical approach to manage gist tokens, which is both conceptually interesting and practically useful.

**Weaknesses**
1. The models used for evaluation (LLaMA-2 and Qwen-2) are relatively outdated compared to more recent LLMs.
2. The RAG scores for QA tasks appear to be quite low, despite prior work [1][2] suggesting that RAG-based methods tend to perform well on QA benchmarks.
3. Some aspects of the experimental setup require further clarification. Several examples are listed in the Questions section below.

[1] Li et. al., Retrieval Augmented Generation or Long-Context LLMs? A Comprehensive Study and Hybrid Approach, EMNLP 2024 Industry Track

[2] Yu et. al., In Defense of RAG in the Era of Long-Context Language Models, arxiv 2024

---

> ### Author Rebuttal · Authors · 2025-07-31
>
> Thank you for your thoughtful evaluation and detailed comments. We address each concern below:
>
> -----
>
> **Q1. “The models used for evaluation (LLaMA‑2 and Qwen‑2) are relatively outdated compared to more recent LLMs.”**
>
>  **Response:**
>
>  As explained in the paper, we use LLaMA‑2‑7B‑Chat because three out of the four compression baselines are based on it, ensuring fair comparisons.
> Our approach is model-agnostic and compatible with any Transformer-based LLM. To further validate its generality, we have extended our experiments to include more advanced and larger models, including Qwen‑2.5‑7B and Qwen‑2.5‑14B. The results are shown in the table below:
>
> | Method                   | Single-Doc QA | Multi-Doc QA | Summarization | Few-Shot | Code |
> | :----------------------- | :------------ | :----------- | :------------ | :------- | :--- |
> | Qwen-2.5-7B              | 41.9          | 45.2         | 26.5          | 69.1     | 64.9 |
> | Qwen-2.5-7B-**FT**       | 42.7          | 46.1         | 26.7          | 67.6     | 66.3 |
> | Qwen-2.5-7B-**Beacon**   | 42.5          | 45.8         | 26.8          | 67.4     | 66.4 |
> | Qwen-2.5-7B-**AdmTree**  | 42.5          | 46.3         | 28.1          | 69.3     | 66.8 |
> | Qwen-2.5-14B             | 42.5          | 52.9         | 25.1          | 71.7     | 66.7 |
> | Qwen-2.5-14B-**FT**      | 43.9          | 50.5         | 27.1          | 68.8     | 67.1 |
> | Qwen-2.5-14B-**Beacon**  | 43.4          | 49.9         | 27.1          | 68.5     | 67.4 |
> | Qwen-2.5-14B-**AdmTree** | 43.5          | 50.8         | 27.5          | 69.7     | 68.1 |
>
> These results confirm that AdmTree continues to perform effectively on larger and more capable models, consistently outperforming the state-of-the-art method Beacon and the strong baseline Original LLM‑FT.
>
> ------
>
> **Q2. “The RAG scores for QA tasks appear to be quite low.”**
>
> **Response:**
>
> Thank you for highlighting the pertinent RAG literature. Our RAG setup follows the original LongBench protocol. We use a chunk size of 500 and retrieve the top‑k most query‑similar chunks to fill the model’s context window as fully as possible, without preserving the original chunk order.
>
> Compared to the paper you mentioned, the relatively low RAG scores in our work mainly stem from two factors. First, we use smaller open-source models, while the cited works rely on much larger and more capable closed-source LLMs such as Gemini‑1.5‑Pro and GPT‑4o. Second, we evaluate on LongBench, while those works use InfiniteBench. These two benchmarks differ substantially and cannot be directly compared. Besides, the original LongBench paper also observed cases where RAG negatively impacted performance.
>
> As shown below, we further explored the effectiveness of the Order-Preserving (OP) RAG method from the paper [1] you referenced, testing it with various embedding models.
>
> |                          | Single-Doc QA | Multi-Doc QA | Summarization | Few-Shot | Code |
> | ------------------------ | ------------- | ------------ | ------------- | -------- | ---- |
> | **LLama-2-7B & vanilla RAG** |               |              |               |          |      |
> | BM25                     | 25.1          | 23.9         | 24.4          | 56.4     | 33.1 |
> | SBERT                    | 17.1          | 15.8         | 23.6          | 53.2     | 36.8 |
> | OpenAI                   | 28.3          | 16.4         | 16.9          | 23.7     | 50.3 |
> | **LLama-2-7B & OP RAG**      |               |              |               |          |      |
> | BM25                     | 22.6          | 21.2         | 15.5          | 55.2     | 30.2 |
> | SBERT                    | 20.3          | 18.9         | 21.2          | 53.6     | 29.1 |
> | SBERT                    | 32.1          | 19.2         | 15.3          | 21.8     | 48.3 |
>
> While OP‑RAG led to modest improvements in some cases, it still failed to outperform AdmTree. Additionally, we observed that RAG offers negative impact in summarization tasks, underscoring its limited generalization across task types. In contrast, AdmTree demonstrates stronger and more consistent generalization across a diverse set of scenarios.
>
> ------
>
> **Q3. “What are the detailed configurations for the RAG experiments?”**
>
> **Response:**
>
> Please refer to the response to Q2.
>
> ------
>
> **Q4. “Why does the original LLM exhibit a speed-up in Table 1 for LLaMA‑2‑7B?”**
>
> **Response:**
>
> The original LLaMA‑2‑7B model has a context window limit of 4096 tokens. Through fine-tuning (Original LLM‑FT), we extended the model’s window length to accommodate the full input from LongBench without truncation. As a result, the original model runs with shorter inputs and thus achieves significantly lower latency compared to the extended model, although its performance is naturally weaker.
>
> ------
>
> **Q5. “What compression ratio was used in Figure 4?”**
>
> **Response:**
>
> In Figure 4, for each context length, we selected the smallest compression ratio from the set {2, 4, 8, 16, 32} to ensure that (context_length / compression_ratio) < model_context_window. We will clarify this in Section 5.3 of the final version.
>
> ------
>
> **Q6. “For the needle-in-a-haystack evaluations, what types of noise and needle content were used? Was the noise content repetitive?”**
>
> **Response:**
>
> We followed the original Needle in a Haystack setup. The noise content comprises essays by Paul Graham, with no repetition. The needle is a synthetic question (to prevent real-world data leakage) about San Francisco.
>
> ------
>
> **Q7. “Limitations were missing from conclusion.”**
>
> **Response:**
>
> Thank you for catching this oversight. We will reintroduce the following limitations in the Conclusion section: (1) Our current experiments are focused on English long-text benchmarks, and the multilingual or cross-lingual capabilities of AdmTree remain unexplored. (2) AdmTree performs compression only on the input side. It does not currently address the compression of long outputs, which is increasingly important as LLMs produce longer intermediate reasoning chains.
>
> ----
>
> We sincerely thank you again for your constructive feedback. We hope that our responses have addressed your concerns and support a higher score!

---

> > ### Comment · Reviewer_eT7L · 2025-08-02
> > **Response to Rebuttal**
> >
> > Thank you for the rebuttal! I appreciate the experiments with more recent LLMs, along with further clarifications and analysis. All concerns are addressed, and I will keep my score to recommend acceptance.

---

> > > ### Author Response · Authors · 2025-08-02
> > >
> > > Thank you for your follow-up and effort. We appreciate your positive feedback and support. Your comments were valuable in helping us improve the paper!

---

### Official Review · Reviewer_FQ4D · 2025-07-03

**Clarity:** 3
**Significance:** 3
**Originality:** 3
**Rating:** 4
**Confidence:** 3

**Summary:**

AdmTree introduces an adaptive, hierarchical compression framework that first segments long input according to information-density, inserts learned “gist” tokens, and then builds a binary semantic tree whose nodes are aggregated with a single-layer self-attention module. Extensive experiments demonstrate the effectiveness of the proposed method.

**Questions:**

See Weaknesses.

**Ethical Concerns:**

["NO or VERY MINOR ethics concerns only"]

**Final Justification:**

The rebuttal satisfactorily addresses my concerns. I look forward to seeing the additional experimental results incorporated into the final version and maintain my recommendation for acceptance.

**Limitations:**

yes

**Paper Formatting Concerns:**

No concerns.

**Quality:**

3

**Strengths And Weaknesses:**

Strengths:
1. The paper is well written and easy to understand.
2. The figures are well plotted to help understand the workload of AdmTree.
3. The proposed compression method looks interesting to me.

Weaknesses:
1. The adaptive token budget is decided solely by an entropy-weighted perplexity score. Have the authors profiled alternative signals—e.g., TF-IDF, mutual information, or a lightweight learned ranker—to confirm the choice is not dataset-specific? An ablation study is required to reveal whether performance gains stem from the tree itself or from the hand-picked heuristic.

2. All internal nodes are compressed by a single self-attention layer followed by pooling. When the hierarchy reaches dozens of levels, can such a shallow module still propagate fine-grained cues?

3. Results cover two 7B chat models on LongBench-style tasks. How does AdmTree fare on code or math benchmarks, or on other model sizes? More experiments on diverse model sizes and domains would demonstrate the robustness and generalization abilities.

---

> ### Author Rebuttal · Authors · 2025-07-31
>
> We sincerely appreciate your recognition of our AdmTree's design and the inspiration your comments brought us. Below, we address the concerns you raised in detail:
>
> ------
>
> **Q1. “The adaptive token budget is decided solely by an entropy-weighted perplexity score. Have the authors considered alternative signals like TF-IDF, mutual information, or a learned ranker?”**
>
> **Response:**
>
>  In our early experiments, we explored several alternative scoring strategies, including attention-based scores and pure perplexity-based ranking, but found them less effective and stable. We have now included additional results comparing different scores:
>
> | Metric                      | Single-Doc QA | Multi-Doc QA | Summarization | Few-Shot | Code |
> | :-------------------------- | :------------ | :----------- | :------------ | :------- | :--- |
> | TF-IDF                      | 35.5          | 35.8         | 25.9          | 64.0     | 59.4 |
> | PPL                         | 35.9          | 35.7         | 26.2          | 65.0     | 59.8 |
> | Entropy                     | 35.8          | 35.2         | 25.9          | 64.1     | 59.7 |
> | Attention Score             | 36.3          | 35.9         | 26.5          | 64.7     | 60.1 |
> | entropy-adjusted PPL (Ours) | 36.5          | 36.3         | 26.9          | 65.5     | 60.9 |
>
> Empirically, our entropy-adjusted PPL shows better alignment with actual information density, since it simultaneously accounts for the difficulty of the sentences and the contextual uncertainty.
>
> Besides, as you suggested, designing a lightweight learnable scorer is indeed a promising direction. For instance, such a scorer could jointly consider the input and the LLM’s own capacity, enabling it to dynamically assign scores that reflect how different LLMs truely perceive the information density of the same sentence.
>
> -----
>
> **Q2. “All internal nodes are compressed by a single self-attention layer followed by pooling. Can such a shallow module still propagate fine-grained cues in deep hierarchies?”**
>
> **Response:**
>
>  The primary function of the aggregator is to propagate semantic information from child nodes in a bottom-up manner. Since it does not need to maintain its own semantic representation, we opted for a lightweight single-layer Transformer to ensure training efficiency.
>
> To further investigate how aggregator depth affects performance at different tree depths, we compared 1-layer and 3-layer aggregators under varying compression ratios (i.e., different tree depths), using Qwen-2-7B as the base LLM. We selected QA and summarization tasks from LongBench, as they require semantic of different levels. LongBench allows inputs up to 32K tokens, so under 2× compression, tree depths can easily exceed 10 layers. The experimental results are as follows:
>
> |                     | 2× Compression Ratio |               | 8× Compression Ratio |               |
> | ------------------- | -------------------- | ------------- | -------------------- | ------------- |
> |                     | Single Doc QA        | Summarization | Single Doc QA        | Summarization |
> | 1-layer aggregators | 42.8                 | 30.5          | 40.8                 | 29.8          |
> | 3-layer aggregators | 43.2                 | 31.3          | 41.3                 | 30.9          |
>
> We observe that deeper aggregators do indeed improve performance, especially in summarization tasks where higher-level semantics are more critical. This suggests that **the current aggregation design still has room for improvement**. Unlike our current shared lightweight aggregators, future work could explore **depth-aware or compression-aware aggregators**. For example, under high compression ratios, leaf nodes must carry more semantic load, whereas in low-compression ratios, deeper trees shift the burden upward, making it essential for aggregators to appropriately balance fine-grained and global semantics.
>
> -----
>
> **Q3. “Results cover two 7B chat models. How does AdmTree perform on code/math tasks or other model sizes?”**
>
> **Response:**
>
> The LongBench benchmark already includes a Code Completion task. In addition, we provided results on LongBench v2 and GSM8K in Tables 4 and 5 of the appendix. LongBench v2 requires long-context reasoning, while GSM8K is a widely used benchmark for mathematical reasoning.
>
> To further demonstrate generalization across model sizes, we extended our experiments to include larger LLMs such as Qwen‑2.5‑7B and Qwen‑2.5‑14B. The results are shown below:
>
> | Method                   | Single-Doc QA | Multi-Doc QA | Summarization | Few-Shot | Code |
> | :----------------------- | :------------ | :----------- | :------------ | :------- | :--- |
> | Qwen-2.5-7B              | 41.9          | 45.2         | 26.5          | 69.1     | 64.9 |
> | Qwen-2.5-7B-**FT**       | 42.7          | 46.1         | 26.7          | 67.6     | 66.3 |
> | Qwen-2.5-7B-**Beacon**   | 42.5          | 45.8         | 26.8          | 67.4     | 66.4 |
> | Qwen-2.5-7B-**AdmTree**  | 42.5          | 46.3         | 28.1          | 69.3     | 66.8 |
> | Qwen-2.5-14B             | 42.5          | 52.9         | 25.1          | 71.7     | 66.7 |
> | Qwen-2.5-14B-**FT**      | 43.9          | 50.5         | 27.1          | 68.8     | 67.1 |
> | Qwen-2.5-14B-**Beacon**  | 43.4          | 49.9         | 27.1          | 68.5     | 67.4 |
> | Qwen-2.5-14B-**AdmTree** | 43.5          | 50.8         | 27.5          | 69.7     | 68.1 |
>
> These results confirm that AdmTree maintains strong performance on larger and more advanced models, consistently outperforming the state-of-the-art method Beacon and the strong baseline Original LLM-FT.
>
> ----
>
> We sincerely thank you again for your thoughtful suggestions and hope our clarifications address your concerns. We would greatly appreciate it if you would consider a higher score.

---

> > ### Comment · Reviewer_FQ4D · 2025-08-04
> >
> > Thank you for the rebuttal, which satisfactorily addresses my concerns. I look forward to seeing the additional experimental results incorporated into the final version, and I will maintain my recommendation for acceptance.

---

> > > ### Author Response · Authors · 2025-08-05
> > >
> > > We are grateful for your insightful comments and feedback! We are committed to incorporating the clarifications and additional experiments discussed during the rebuttal to further enhance the quality of our work.

---

### Official Review · Reviewer_68Wb · 2025-07-07

**Clarity:** 3
**Significance:** 4
**Originality:** 4
**Rating:** 5
**Confidence:** 4

**Summary:**

This paper deals with long contexts in LLM. Essentially, the proposed AdmTree incrementally builds a hierarchical summary as a binary tree of gists to leverage it for the prediction of the next word in a framework of causal language model. As described in Section 3.1, it dynamically segments input context, assigns gist tokens summarize them, and builds a tree structure to capture a long context. It consists of three stages, namely dynamic gist token constuction, tree construction, and tree-based compression for prediction while freezing original LLM.
Extensive experiments show that the proposed AdmTree has much better accuracy for prediction, and the model is further analyzed from various viewpoints to show how the AdmTree actually works.

**Questions:**

- As described in Section 3.3, AdmTree utilizes self-attention from subsequent part of texts, mitigating the unidirectional limitation of causal LLM. This is conceptually good, but I wonder it might influence the causal structure as a probabilistic model, namely it could be somewhat cheating for predicting the next word. Is this true or not?
- Semantic density of text segment is measured by Equation (2) using a combination of perplexity and entropy. However, perplexity and entropy are actually the same thing, just differing about some transformation. What is occurring here?
- This is not a question, but it is a bit pity that the hierarchically organized gists are just flattened for  inherently flat attention mechanism of current LLM. If LLM itself utilizes this kind of hierarchical structure for prediction, depth-first flattening might be no longer necessary.

**Ethical Concerns:**

["NO or VERY MINOR ethics concerns only"]

**Limitations:**

Nothing.

**Paper Formatting Concerns:**

Nothing.

**Quality:**

4

**Strengths And Weaknesses:**

This is a carefully built, and scientifically interesting model for intrinsic summarization for LLM objective. Hiearchical organization of gists is shown to be a crucial component for this model, and semantic heterogeneity of a text stream is also considered decently.
Technically speaking, I am not perfectly sure that the proposed model is completely causal (as described below), but this is ia minor problem and this paper is clearly worth being published.

---

> ### Author Rebuttal · Authors · 2025-07-31
>
> We sincerely thank you for the detailed and thoughtful review, as well as the recognition of our work. We address your comments and questions below:
>
> -----
>
> **Q1. On causal structure and use of self-attention from subsequent segments**
>
> > *“AdmTree utilizes self-attention from subsequent part of texts, mitigating the unidirectional limitation of causal LLM... it could be somewhat cheating for predicting the next word.”*
>
> **Response:**
>
>  Thank you for pointing out this subtle but important concern. To clarify, AdmTree preserves the causal structure required for next-token prediction. The self-attention involved in tree aggregation (Section 3.3) operates **outside the autoregressive decoding path**. It only summarizes the **already-seen** gist tokens to build semantic tree. During next-token prediction in segment $s_k$, the model only attends to tokens from $s_k$ (autoregressively, via causal masking), and the semantic tree $T_{<k}$, constructed solely from previous sub-segments. Thus, no future tokens beyond position $j$ (the predicted token) are accessed or leaked into the prediction path.
>
> -----
>
> **Q2. On the use of both perplexity and entropy in Equation (2)**
>
> > *“Perplexity and entropy are actually the same thing, just differing about some transformation.”*
>
> **Response:**
>
> Perplexity and entropy would be mathematically equivalent only when entropy were computed as the cross-entropy loss on the target token. But in our framework, they serve different roles. For segment $X_i = [x_1, \dots, x_n]$, PPL measures the model's sequence-level prediction difficulty, which is defined as: $\text{PPL}(X_i) = \exp\left(-\frac{1}{n} \sum_{t=1}^{n} \log P(x_t \mid x_{<t})\right)$
> This reflects how “surprised” the model is by the segment. In contrast, Entropy in our design is the average token-level entropy, which is calculated as: $\text{Entropy}(X_i) = \frac{1}{n} \sum_{t=1}^{n} \left(-\sum_{v \in \mathcal{V}} P(v \mid x_{<t}) \log P(v \mid x_{<t})\right)$
> where $\mathcal{V}$ is the LLM vocabulary. This captures the intrinsic contextual complexity at each position. High PPL may arise from genuine surprising content or from low-value noise. The entropy term helps suppress segments that are surprising but semantically flat, avoiding over-allocation of gist tokens to noise. Empirically, our entropy-adjusted PPL shows better alignment with actual information density, especially in heterogeneous contexts.
>
> We also conducted comparative experiments showing the superiority of our entropy‑adjusted PPL metric:
>
> | Metric                      | Single-Doc QA | Multi-Doc QA | Summarization | Few-Shot | Code |
> | :-------------------------- | :------------ | :----------- | :------------ | :------- | :--- |
> | TF-IDF                      | 35.5          | 35.8         | 25.9          | 64.0     | 59.4 |
> | PPL                         | 35.9          | 35.7         | 26.2          | 65.0     | 59.8 |
> | Entropy                     | 35.8          | 35.2         | 25.9          | 64.1     | 59.7 |
> | Attention Score             | 36.3          | 35.9         | 26.5          | 64.7     | 60.1 |
> | entropy-adjusted PPL (Ours) | 36.5          | 36.3         | 26.9          | 65.5     | 60.9 |
>
> ----
>
> **Q3. On flattening the hierarchical structure before feeding into flat LLMs**
>
> **Response:**
>
> > *“It is a bit pity that the hierarchically organized gists are just flattened... If LLM itself utilizes hierarchical structure, flattening might be unnecessary.”*
>
> This is an insightful comment and aligns with our future vision! We currently flatten the hierarchical gist tree to maintain compatibility with Transformer-based architectures, but this is a practical constraint, not a conceptual necessity.
>
> However, emerging architectures like **Hierarchical Reasoning Model** [1], which employs separate recurrent modules for planning and execution while using very few parameters, perform deep hierarchical reasoning without any flattening and still achieve state-of-the-art results. Similarly, recent research into **diffusion-based reasoning**, such as Diffusion-of-Thought (DoT) [2], allows reasoning steps to form non-linear, chain-of-thought-like trajectories during diffusion rather than a strict left‑to‑right generation.
>
> Together, these models illustrate that hierarchical representations can be processed natively. In future work, AdmTree’s semantic tree could serve as a compression front-end feeding into these hierarchy-native backbones, thereby eliminating the need for flattening and enabling more natural hierarchical reasoning. We will add this forward-looking discussion in our conclusion section.
>
> *[1] Hierarchical Reasoning Model. https://github.com/sapientinc/HRM*
>
> *[2] Diffusion of thought: Chain-of-thought reasoning in diffusion language models. NIPS'24.*
>
> ----
>
> Thank you for your valuable insights and kind recognition. Your feedback has inspired us and made our work better!

---

### Author Response · Authors · 2025-08-09
**Rebuttal Summary and Appreciation to AC and Reviewers**

Dear AC and Reviewers,

As the rebuttal deadline approaches, we sincerely thank all reviewers for your insightful feedback and the AC for your help, as well as for the consistent recognition of our work.

In this paper, we proposed the hierarchical AdmTree framework, which achieves efficient compression of lengthy text while preserving semantic information in a more comprehensive manner. Experiments across various compression scenarios, presented in both the main paper and the appendix, demonstrate that AdmTree significantly outperforms existing state-of-the-art models.

During the rebuttal period, we addressed your concerns through additional experiments and clarifications, including performance on larger LLM backbones (FQ4D, eT7L), the choice of the information density metric (68Wb, FQ4D), and comparisons with more advanced RAG-based methods (eT7L, LLL7).

**We are glad to have resolved concerns from all four reviewers, and we promise to incorporate these results and discussions into the final version**. Thank you again for helping us improve our work.


Wishing you all the best,

Authors of Paper 24435

---

### Decision · Program_Chairs · 2025-09-17

**Decision:**

Accept (poster)

**Comment:**

***Summary***

The authors introduce AdmTree, which is a mechanism for compressing the KV cache, and thus reducing the cost of attention in LLMs.  Building on prior published work, AdmTree first inserts *gist tokens* at equally spaced intervals in the input text.  The gist tokens attend to and summarize the input text.  Later tokens (in more distant segments), then attend to the gist tokens rather than attending directly over the entire sequence length.

The key innovation of AdmTree is that instead of simply inserting gist tokens at regularly spaced intervals, it constructs a binary tree of gist tokens, where parents attend to children, and children attend to the input text.  The binary tree is then flattened to produce a KV-cache that other text segments can attend to.  In addition, AdmTree attempts to allocate gist tokens dynamically, by sorting segments according to a custom metric that includes average token entropy, so that more tokens are allocated to segments with greater information content.

The authors compare against various baselines including RAG, and on various tasks and datasets, and show substantial improvements over SOTA.

***Meta-review***

The reviews for this paper were mostly positive, and all in the range of "borderline accept": (4, 4, 4, 5).  The reviewers asked various clarifying questions during the discussion period, and the authors gave detailed and well thought-out responses.  The authors also ran additional experiments in response to the reviewer questions.  At the end of the discussion period, all of the reviewers stated that their concerns had been addressed.

I have have gone over the paper myself, and I agree with the reviewers.  This paper lies somewhere between "borderline accept" and "accept".  To be honest, the basic idea is not very exciting in my opinion; it is an incremental improvement over prior work.  On the other hand, I can find no grounds to reject the paper either; the literature review is good, the experiments are solid and comprehensive, and my own questions were answered during the rebuttal period.

I am recommending "accept" mainly on the strength of the author rebuttal.  The authors worked hard to address all of the reviewer concerns, and the reviewers in turn stated that all of their concerns had been addressed.  Here's a summary of the discussion.

**Reviewer 68Wb (rating 5)**:
* Voiced concern about whether the binary tree breaks causality.  (This was also a question that I had when reading the paper).  The authors gave a satisfactory explanation... causality is preserved.
* Asked about alternatives to entropy.  The authors provided a detailed table of alternatives that they tested.

**Reviewer FQ4D (rating 4)**:
* Also asked about entropy, and received the same response as reviewer 68Wb.
* Wanted additional tests with bigger models on code and math benchmarks.  The authors ran additional experiments, as requested.

**Reviewer ET7L (rating 4)**:
* Wanted to know more about the RAG comparisons, which the authors answered.  The reviewer stated that their concerns were addressed.

**Reviewer LLL7 (rating 4)** writes:
* "While [the use of a frozen pre-trained LLM] makes the approach model-agnostic and easy to integrate with any LLM, the model itself is unaware of or unoptimized for the compression mechanism."  The authors gave a detailed discussion, and the reviewer was satisfied.